# SCPILOT: Large Language Model Reasoning Toward Automated Single-Cell Analysis and Discovery

**Yiming Gao**[14*†]**, Zhen Wang**[12*‡]**, Jefferson Chen**[1]**, Mark Antkowiak**[1]**, Mengzhou Hu**[1]**,
**JungHo Kong**[1]**, Dexter Pratt**[1]**, Jieyuan Liu**[1]**, Enze Ma**[1]**, Zhiting Hu**[1]**, Eric P. Xing**[23]

[1]UC San Diego, [2]MBZUAI, [3]CMU, [4]Texas A&M
yiminggao618@tamu.edu, zhw085@ucsd.edu

## Abstract

We present SCPILOT, the first systematic framework to practice *omics-native reasoning*: a large language model (LLM) converses in natural language while directly inspecting single-cell RNA-seq data and on-demand bioinformatics tools. SCPILOT converts core single-cell analyses, i.e., cell-type annotation, developmental-trajectory reconstruction, and transcription-factor targeting, into step-by-step reasoning problems that the model must solve, justify, and, when needed, revise with new evidence. To measure progress, we release SCBENCH, a suite of 9 expertly curated datasets and graders that faithfully evaluate the omics-native reasoning capability of SCPILOT w.r.t various LLMs. Experiments with o1 show that *iterative* omics-native reasoning lifts average accuracy by 11% for cell-type annotation and `Gemini-2.5-Pro` cuts trajectory graph-edit distance by 30% versus one-shot prompting, while generating transparent reasoning traces explain marker gene ambiguity and regulatory logic. By grounding LLMs in raw omics data, SCPILOT enables auditable, interpretable, and diagnostically informative single-cell analyses. [4]

## 1 Introduction

In the era of exponential growth in biological data, the quest for artificial intelligence (AI) that can function as a true scientific assistant to automate and interpret complex scientific analyses has never been more urgent. Recently, large language models (LLMs) have demonstrated surprising breadth in factual recall and reasoning prowess [73, 26, 32, 22], prompting the question: *Can LLMs be harnessed as **genuine scientific partners** to revolutionize traditional biological discovery pipeline?*

Yet, translating these general LLM capabilities into the realm of single-cell biology remains challenging. The surge of single-cell omics has shifted biology from bulk averages to million-cell matrices [63, 9, 14], but analysis pipelines still depend on implicit, human-only reasoning [39, 54, 17] (Figure 1). While LLMs excel at natural-language explanation and reasoning, most current uses in computational biology utilize LLMs simply as interfaces that invoke existing bioinformatics tools [75, 11, 31], relying solely on these tools' inherent functionalities. Other approaches heavily train foundation models to embed single-cell counts into opaque, high-dimensional vector spaces [77, 15, 67], resulting in less interpretable analyses critical to biological discovery.

We propose to bridge this gap with *omics-native reasoning (ONR)*—a new interactive paradigm in which an LLM (i) receives a concise textual summary derived from the single-cell expression matrix, (ii) explicitly articulates biological hypotheses in natural language, (iii) invokes targeted

---

*Equal contribution

†Work done during his time at UCSD

‡Corresponding author

[4]Code, data, and package are available at `https://github.com/maitrix-org/scPilot`

bioinformatics operations directly on the raw data, (iv) evaluates and interprets numerical evidence, and (v) iteratively refines its reasoning until arriving at biologically coherent conclusions. As shown in Figure 1, by closely coupling reasoning to *raw omics data*, ONR generates transparent and auditable analyses, facilitating interpretability, scientific rigor, and human validation.

This paper operationalizes ONR through SCPILOT, a systematic framework that harnesses the reasoning capabilities of an off-the-shelf LLM integrated with a problem-to-text converter and a curated bioinformatics tool library. SCPILOT explicitly formulates and iteratively refines hypotheses, addressing three canonical single-cell challenges: *cell-type annotation*, *developmental-trajectory reconstruction*, and *transcription-factor targeting*, producing transparent and biologically insightful reasoning processes. To systematically quantify

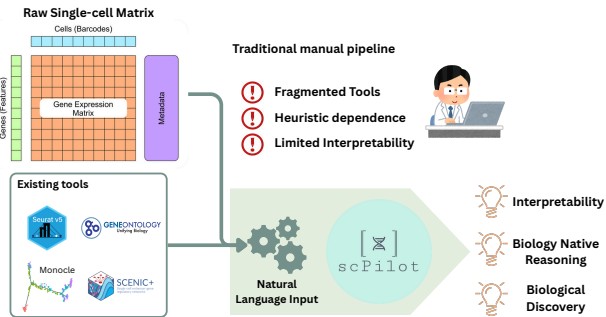

Figure 1: Human-like reasoning + established bioinformatics tools = hands-free single cell analysis

progress, we further introduce SCBENCH, the first benchmark for omics-native reasoning that scores numerical accuracy and reveals the biological validity of the model's narrative across nine expertly curated single-cell tasks. Our contributions are fourfold:

- **LLM-driven single-cell analysis framework**. We formulate the first *omics-native reasoning*, language-centric workflow that automates key analytic stages—cell-type annotation, trajectory inference, and gene-regulatory network prediction—while preserving scientific transparency.

- **Comprehensive benchmark suit**. SCBENCH offers task-specific metrics and expert-verified ground truth, enabling objective comparison of LLMs on biologically meaningful problems.

- **Empirical insights and validation.** Comprehensive experiments across nine benchmark datasets demonstrate the effectiveness of SCPILOT: iterative omics-native reasoning lifts average cell-type annotation accuracy by 11%, reduces trajectory graph-edit distance by 26%, and improves GRN prediction AUROC by 0.03 over direct prompting and conventional baselines.

- **Biological interpretability and diagnostic reasoning.** SCPILOT generates transparent reasoning traces that expose marker ambiguities, lineage inconsistencies, and tissue-specific regulatory logic, enabling biologically interpretable and diagnostically informative single-cell analyses.

## 2 Related Work

**Large Language Models in Single-Cell Analysis.** Early biomedical LLMs, e.g., BioGPT [48], BioMedLM [8], and Galactica [66], showed that pre-training on PubMed abstracts or full-text markedly improves factual recall and zero-shot QA, while newer general LLMs (e.g., GPT-4o, Claude-3) now rival or exceed them with broader literature coverage. In parallel, a growing family of single-cell foundation models [77, 21, 15, 67, 59, 25, 58, 41, 6, 65, 37], mostly encoder-style LLMs that treat genes as tokens to learn gene- and cell-level embeddings for imputation, perturbation prediction, and cross-dataset transfer. Cell2Sentence and C2S-Scale [41, 56] encode each cell as a "sentence," enabling natural-language queries, while other works build LLM interfaces for single-cell data via fine-tuning [46, 61, 42] or autonomous tool agents [27, 19, 57, 75, 11]. General-purpose biomedical agents such as Biomni [31] demonstrate autonomous problem-solving across domains.

Despite their progress, these approaches sidestep the core cognitive load of single-cell analysis: embedding models speak in vectors with no explanations, chat wrappers and tool agents re-package fixed results from traditional tools. Yet we need more language-native reasoning for single-cell analysis, the ability for an LLM to argue, justify, and iteratively refine biological conclusions. Prior work [28] showed GPT-4 can label cell types directly from marker genes. SCPILOT pushes this paradigm beyond a single downstream task to the entire analytic workflow systematically.

**Automated Single-Cell Analysis Pipelines.** Modern single-cell workflows rely on comprehensive toolkits like Seurat and Scanpy [74, 62] as the backbone. Specialized modules like CellTypist (cell-type annotation), Monocle (developmental trajectory reconstruction), and SCENIC (gene regulatory

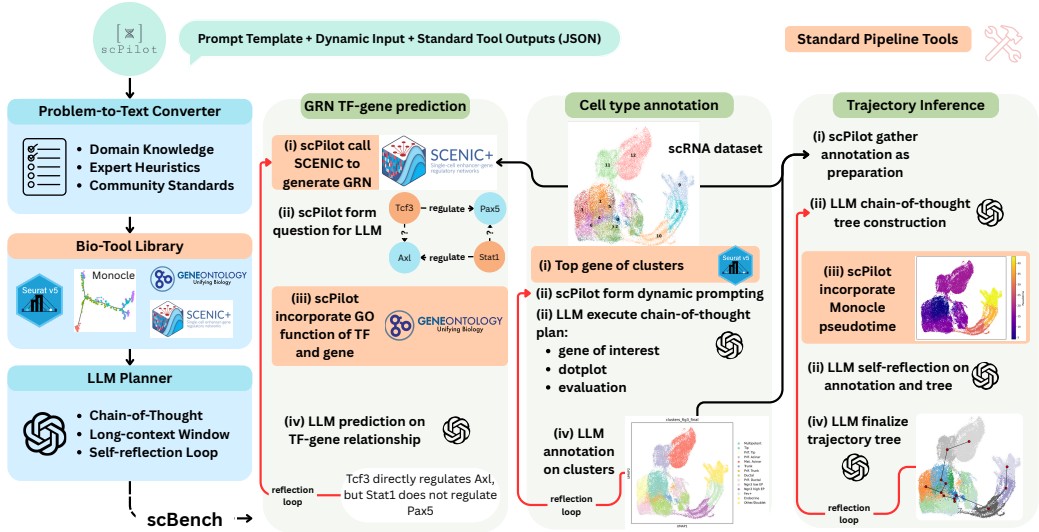

Figure 2: Overview of the SCPILOT framework. The system integrates a problem-to-text converter, an LLM planner, and a bio-tool library to perform iterative reasoning and tool calls for three workflows: cell-type annotation, trajectory inference, and gene-regulatory prediction.

networks) [16, 68, 4] address specific subtasks but expose many hyperparameters and opaque defaults. Web-based platforms (e.g., ASAP [20]) and one-click frameworks (e.g., SPEEDI [71]) simplify execution yet still embed rigid heuristics that may fail on new tissues or perturbations. Recent LLM tool agents ease this burden by writing code and invoking domain packages on demand [27, 19, 64, 11, 57]. CellAgent [75], for example, uses GPT-4 to automatically select tools and hyperparameters. While impressive, these systems mainly wrap default heuristics and offer limited biological insights behind tool calling. SCPILOT advances from scripted automation to *co-piloting*: not only to call tools, the model needs to interpret the output of Monocle or SCENIC and perform profound biological reasoning for discovery.

**Benchmarks for Biological Reasoning** Existing benchmarks for single-cell foundation models or algorithms emphasize embedding quality or numeric metrics [44, 7, 47, 60], offering little transparency for biologically meaningful interpretation [35, 70, 7]. LLM benchmarks such as BioASQ [38], PubMedQA [34], MedQA-USMLE [33], and recent GPQA [55] and LAB-Bench [40] test factual recall but not operation on raw omics data. More recent agent-centric suites (BixBench [51], ScienceAgentBench [12], FIRE-Bench [72]) test LLMs to orchestrate code execution across natural language bioinformatics problems, but their coverage is shallow, and their evaluation is without domain-deep omics reasoning. Isolated efforts have probed language-native biology, e.g., GPT-4 for cell-type annotation [28] and LLMs for gene set function discovery [30], yet no public benchmark covers multiple single-cell workflows with ground-truth answers and demands explicit biological justification. To fill this gap, our proposed SCBENCH systematically evaluates language-native reasoning across multiple single-cell workflows with curated datasets, numeric ground truth, and automatic metrics—providing the first rigorous benchmark for the co-pilot paradigm in SCPILOT.

## 3   SCPILOT: Automation of Single-Cell Analysis by LLMs

Let $\mathbf{X} \in \mathbb{R}^{G \times N}$ be a single-cell expression matrix with $G$ genes and $N$ cells, and let $\boldsymbol{q}$ denote a biological query (e.g. "What is the cell type of cluster 5?" or "Does TF $Z$ regulate gene $Y$?").

Classical bioinformatics methods typically address query $\boldsymbol{q}$ by executing a predetermined pipeline:

$$\widehat{\boldsymbol{y}} \;=\; g_{\text{tool}}\big(\mathbf{X}; \theta\big), \tag{1}$$

where $g_{\text{tool}}$ is selected from established bioinformatics tools such as *Scanpy* or *Monocle*, and $\theta$ is a set of hyperparameters manually tuned based on implicit biological assumptions and analyst expertise. Although effective, this traditional practice obscures the underlying biological rationale behind the chosen parameters, limiting reproducibility, transparency, and interpretability.

Recent LLM-based "tool agents" tackle bioinformatics tasks by writing code on the user's behalf: the model produces a Python or R snippet, executes it, prints the numeric output, and (optionally) summarizes the result in text. Formally, given a query $q$ and an expression matrix $\mathbf{X} \in \mathbb{R}^{G \times N}$, the agent generates code $\texttt{src}_1, \ldots, \texttt{src}_K$ such that

$$\widehat{\boldsymbol{y}} = g_{\texttt{src}_K}\big(g_{\texttt{src}_{K-1}}(\ldots g_{\texttt{src}_1}(\mathbf{X})\ldots)\big), \tag{2}$$

where each function $g_{\texttt{src}_k}$ corresponds to an invoked bioinformatics operator (e.g., clustering via $\texttt{scanpy.tl.leiden}$). The model's *reasoning lives mainly in comments and chat text* surrounding the code; intermediate raw data and numerical results are hidden unless explicitly printed. Consequently, the human analyst must read both Python and text to audit a run, and the causal logic link between numeric evidence and biological claims is easily lost.

**Omics-Native Reasoning (ONR).** We instead require the LLM to reason *directly over the omics space* and to record every claim, evidence pair in a transparent trace. Let $S_0 = \mathbf{X}$. At step $k$, the reasoner emits a pair $(c_k, o_k)$,

$$S_k = o_k(S_{k-1}), \tag{3}$$

where $c_k$ is a natural-language claim, justification, or decision, and $o_k$ is a *primitive omics operator*, a single, verifiable action applied to the current data state $S_{k-1}$ (e.g. filtering, clustering, scoring, look-ups, etc). The sequence

$$\mathcal{R} = \big[(c_1, o_1), \ldots, (c_K, o_K)\big] \tag{4}$$

constitutes a *verbal + computational proof*. The final state $S_K$ is mapped to a prediction $\widehat{\boldsymbol{y}} = h(S_K)$ that answers $q$. The final evaluation is conducted by comparing $\widehat{\boldsymbol{y}}$ with a ground truth answer $\boldsymbol{y}$.

### 3.1 The SCPILOT Framework

To operationalize *omics-native reasoning* for large-scale single-cell experiments, we introduce SCPILOT, a modular framework that systematically transforms high-dimensional omics data into concise textual summaries, guides reasoning via LLMs, and selectively invokes computational biology tools to iteratively gather and assess evidence. Formally, given a biological query $q$ and an expression matrix $\mathbf{X}$, SCPILOT produces a prediction $\hat{\boldsymbol{y}}$ alongside a transparent reasoning trace $\mathcal{R}$. The architecture comprises three interacting components (Figure 2):

**Problem-to-Text Converter** $\mathcal{C}$. Single-cell datasets routinely contain $N \sim 10^{5-6}$ cells, far beyond any context window of current LLMs. Thus, for each query $q$, we implement an algorithmic mapping:

$$\boldsymbol{\Phi}_q : \mathbb{R}^{G \times N} \longrightarrow \mathcal{S}_q, \tag{5}$$

which produces a *semantic sketch* $\boldsymbol{s}_q = \boldsymbol{\Phi}_q(\boldsymbol{X})$ digestible by the LLM within a single prompt. The map is *algorithmic, not learned* (examples presented in Table 1). Crucially, $\boldsymbol{\Phi}_q$ reduces volume while preserving biological salience, (e.g., reporting cluster sizes, top-ranked marker genes, developmental trajectory connections, or transcription factor-target scores) rather than presenting the full expression matrix, thus significantly reducing data dimensionality while retaining critical biological context.

**Bio-Tool Library** $\mathcal{T}$. This curated library provides a set of primitive omics operators $o_k \in \Omega$ that encapsulate well-established bioinformatics routines (e.g., *Scanpy*, *Seurat via Reticulate*, *Monocle 3*, *pySCENIC*), and lightweight plotting utilities. Each operator returns structured, machine-parsable JSON outputs (e.g., numeric scores, ranked gene lists, graphs) accompanied by a succinct natural-language description, enabling the reasoner to seamlessly integrate fresh computational evidence into subsequent reasoning steps in $c_{k+1}$.

**LLM Reasoner** $\mathcal{R}_\phi$. The reasoning module $\mathcal{R}_\phi$, instantiated by powerful LLMs (such as o1), receives the textual summary, user-defined biological query, and a structured *reasoning scaffold* refined through iterative prompting and domain-expert heuristics. These elements jointly establish a closed-loop reasoning workflow:

$$\mathbf{X} \xrightarrow{\mathcal{C}} \text{Prompt} \xrightarrow{\mathcal{R}_\phi} \{\text{Thought}_k, \text{Call}_k\}_{k=1}^K \xrightarrow{\mathcal{T}} \mathcal{R}_{1:K} \longrightarrow \hat{\boldsymbol{y}}. \tag{6}$$

**Design Principles.** SCPILOT provides a modular, flexible blueprint that enables researchers to customize reasoning workflows tailored explicitly to their biological queries. The framework adheres to three rigorously validated design principles essential for achieving optimal performance: (a) *Biological context first*: Prompts consistently incorporate key biological metadata, such as species, tissue

Table 1: Summary of SCBENCH. Each row defines a computational task, how it's compressed into text, datasets used for evaluation, metrics for assessment, and ground truth sources.

| Task | Compression (Problem → Text) | Datasets | Metric | Ground truth |
|---|---|---|---|---|
| **Cell-type annotation** | Scanpy–Leiden clusters + top-$k$ marker genes per cluster ($k = 10$) | PBMC3k [1], Liver [43], Retina [50] | Cluster-level accuracy (1 / 0.5 / 0) [28] | Author-provided labels |
| **Trajectory inference** | PAGA or Monocle-3 lineage graph and pseudotime; landmark genes varying monotonically | Pancreas [5], Liver [45], Neocortex [53] | Node-Jaccard ↑, Graph-Edit ↓, Spectral Dist. ↓ | Human-curated lineage tree in original study |
| **GRN edge prediction** | Top-$M$ TF–gene pairs from pySCENIC with motif support ($M \leq 150$) | GRNdb stomach, liver, kidney [18] + TRRUST validation [24] | AUROC | Experiment-validated edges from TRRUST v2 [24] |

type, and experimental protocol. Expert knowledge is required to choose the context, from previous reasoning or tool calls. (b) *Iterative reasoning*: Reasoning and reflection are unfolded iteratively, systematically refining hypotheses based on accumulating computational evidence and even pervious mistakes. (c) *Minimal manual heuristics*: We seed each task with high-level prompts distilled from domain best practices, without task-specific fine-tuning of LLM parameters; performance improvements arise exclusively through enhanced prompting strategies and richer evidence.

## 3.2 SCBENCH: Benchmarking SCPILOT with Real-World Biological Meaningful Tasks

We introduce SCBENCH (summarized in Table 1), a comprehensive benchmark designed explicitly to evaluate the biological reasoning capabilities of SCPILOT across representative single-cell RNA sequencing (scRNA-seq) analysis tasks. These tasks encapsulate the analytical complexity and experimental challenges commonly encountered in practical single-cell studies. Datasets included in SCBENCH are carefully selected from high-quality, publicly available scRNA-seq studies, providing a realistic and diverse platform for assessing SCPILOT 's utility and robustness in bioinformatics discovery. To ensure fairness, reproducibility, and rigorous evaluation, termination conditions for each task within SCBENCH are pre-specified rather than being autonomously determined by the LLM.

**Cell Type Annotation.** Given a scRNA expression matrix, the goal of this foundational task is to assign biologically accurate cell-type labels to each cell. Traditionally, this has relied heavily on manual annotation due to limitations in automated tools. We thus curated manually annotated scRNA datasets from published papers: PBMC3k dataset [1] from 10x Genomics, Liver [43], and Retina [50]. SCPILOT employs a fixed maximum of three reasoning iterations, providing the LLM sufficient scope to iteratively refine hypotheses and self-correct without "overthinking" or excessive computational cost.

Table 2: Cell type annotation scores across datasets. Values represent mean ± SD where available. Higher values indicate better performance. The top three performances for each column are highlighted with decreasing background intensity.

| Method | Liver | PBMC | Retina |
|---|---|---|---|
| CellTypist | 0.464 | 0.563 | 0.388 |
| GPTCellType | 0.404 ± 0.141 | 0.613 ± 0.228 | 0.300 ± 0.297 |
| CellMarker 2.0 | 0.304 | 0.250 | 0.632 |
| Biomni (Gemini-2.5 Pro) | 0.464 ± 0.047 | 0.646 ± 0.095 | 0.570 ± 0.135 |
| Direct (o1) | 0.560 ± 0.032 | 0.667 ± 0.071 | 0.474 ± 0.045 |
| SCPILOT (o1) | 0.518 ± 0.032 | 0.792 ± 0.071 | 0.728 ± 0.084 |

**Trajectory Inference.** This task involves reconstructing cellular developmental progression paths, typically structured as lineage trees. Conventionally, trajectory reconstruction relies on statistical tools such as *Monocle* [10] and manual validation by domain experts. We selected three scRNA-seq datasets adhering to stringent criteria: 1) datasets representing clear cellular differentiation processes, 2) original studies that explicitly included trajectory analysis, and 3) availability of expert-curated trajectory lineages as ground truth. The selected datasets are Pancreas [5], Liver [45], and Neocortex [53]. This task is implemented as a single-pass reasoning process, incorporating an initial trajectory construction followed by a controlled refinement step guided by *Monocle*'s output.

**Gene Regulatory Network Prediction.** Given a transcription factor (TF) and target gene pair, the task is to predict the existence of a regulatory relationship. Standard approaches typically utilize computational pipelines like *SCENIC* [4] for candidate predictions, subsequently validated through laboratory experiments and consolidated in reference databases such as TRRUST [24]. For

Table 3: Cell Type Annotation Performance across Different LLMs and Datasets. Values represent performance metrics with standard deviations. The top three performances for each dataset and task type are highlighted with decreasing background intensity.

| LLM | PBMC3k | | Liver | | Retina | |
|---|---|---|---|---|---|---|
| | **Direct** | **SCPILOT** | **Direct** | **SCPILOT** | **Direct** | **SCPILOT** |
| GPT-4o | $0.604_{\pm 0.005}$ | $0.646_{\pm 0.017}$ | $0.440_{\pm 0.002}$ | $0.512_{\pm 0.002}$ | $0.439_{\pm 0.002}$ | $0.675_{\pm 0.011}$ |
| GPT-4o-mini | $0.625_{\pm 0.000}$ | $0.604_{\pm 0.001}$ | $0.339_{\pm 0.003}$ | $0.387_{\pm 0.004}$ | $0.404_{\pm 0.000}$ | $0.447_{\pm 0.026}$ |
| O1 | $0.667_{\pm 0.005}$ | $0.792_{\pm 0.005}$ | $0.560_{\pm 0.001}$ | $0.518_{\pm 0.001}$ | $0.474_{\pm 0.002}$ | $0.728_{\pm 0.007}$ |
| O1-mini | $0.646_{\pm 0.009}$ | $0.521_{\pm 0.001}$ | $0.351_{\pm 0.000}$ | $0.435_{\pm 0.005}$ | $0.456_{\pm 0.001}$ | $0.649_{\pm 0.003}$ |
| Gemini 2.0 Pro | $0.604_{\pm 0.005}$ | $0.792_{\pm 0.009}$ | $0.494_{\pm 0.001}$ | $0.509_{\pm 0.008}$ | $0.491_{\pm 0.000}$ | $0.763$ |
| Gemini 2.0 Flash | $0.604_{\pm 0.001}$ | $0.500_{\pm 0.004}$ | $0.411_{\pm 0.000}$ | $0.435_{\pm 0.001}$ | $0.500_{\pm 0.001}$ | $0.623_{\pm 0.006}$ |
| Gemini 2.5 Pro | $0.583_{\pm 0.001}$ | $0.708_{\pm 0.021}$ | $0.494_{\pm 0.007}$ | $0.488_{\pm 0.001}$ | $0.482_{\pm 0.001}$ | $0.675_{\pm 0.003}$ |
| Gemma 3 27B | $0.479_{\pm 0.001}$ | $0.500_{\pm 0.016}$ | $0.345_{\pm 0.000}$ | $0.393_{\pm 0.004}$ | $0.526_{\pm 0.005}$ | $0.579_{\pm 0.002}$ |

benchmarking, we compiled GRN data from GRNdb [18], a comprehensive database containing TF-gene predictions derived from omics data via SCENIC. We selected three representative tissues from *GRNdb—Stomach, Liver, and Kidney*—and incorporated experimentally validated TF-gene pairs from TRRUST as ground truth. This task is structured as a single-pass reasoning exercise, with all relevant evidence presented upfront for a thorough, integrated reasoning process.

# 4 Experiments

**Benchmarked Models.** We evaluated eight models, including seven proprietary and one prominent open-source model, to represent diverse performance tiers and availability. Proprietary models include `GPT-4o`, `GPT-4o-mini`, `o1`, `o1-mini` [52], `Gemini-2.0-Pro`, `Gemini-2.0-Flash-Thinking`, and `Gemini-2.5-Pro` [13]. The open-source one is `Gemma-3-27B` [3], among the best available at the time of experimentation. To facilitate direct comparison, each primary model was evaluated alongside its lightweight variant (e.g., `-mini`). Further details regarding model versions are provided in the Supplementary Material.

**Baseline Methods.** To rigorously assess performance, we benchmarked SCPILOT against relevant baseline methods across traditional bioinformatic methods and recent LLM-based methods.

*Cell-Type Annotation*: Four established baseline approaches were included—traditional machine learning and database-driven methods (*Celltypist 1.7.1* [76], *CellMarker 2.0* [29]), and LLM-based methods (*GPTCelltype* [28], Biomni [31]).

*Trajectory Inference*: Two baseline methods were utilized—*py-Monocle* [10], a conventional trajectory inference method, and Biomni, representing an advanced LLM-driven pipeline.

*Gene Regulatory Network (GRN) Prediction*: We implemented and evaluated three graph neural network architectures—Graph Convolutional Networks (GCN) [36], Graph Attention Networks (GAT) [69], and GraphSAGE [23]—each trained on the GRNdb dataset. Additionally, we compared our approach with two contemporary LLM-based methods: *LLM4GRN* [2] and *BioGPT* [49].

*Direct Prompting*. To further contextualize SCPILOT's performance, we implemented straightforward *direct* LLM-prompting baselines for each task. All prompts are provided in Appendix.

- *Cell-type annotation*: A single LLM call directly assigns cell-type labels based on differentially expressed genes per cluster, supplemented by high-level dataset descriptions.
- *Trajectory inference*: Consists of three sequential, independent LLM calls: initial cell-type annotation, subsequent trajectory inference using the annotations, and a joint reconsideration step informed by *py-Monocle* results.
- *GRN prediction*: A single-step LLM prompt directly predicts TF-gene regulatory relationships without iterative refinement.

## 4.1 Main Result Analysis

**Cell-type Annotation.** Table 2 demonstrates that both *direct* prompting and the SCPILOT framework outperform traditional annotation tools. Table 3 summarizes the mean accuracy $\pm$ variance across

Table 4: Trajectory reconstruction performance across different LLMs. Values represent mean $\pm$ standard deviation. For Jaccard, higher values ($\uparrow$) are better; for GED-nx (10s) and Spectral Distance, lower values ($\downarrow$) are better. The top three performances for each column are highlighted with decreasing background intensity.

| LLM | Pancreas | | Liver | | Neocortex | |
|---|---|---|---|---|---|---|
| | **Direct** | **SCPILOT** | **Direct** | **SCPILOT** | **Direct** | **SCPILOT** |
| **Jaccard ($\uparrow$)** | | | | | | |
| GPT-4o | $0.923 \pm 0.077$ | $0.872 \pm 0.118$ | $0.956 \pm 0.032$ | $0.978 \pm 0.032$ | $0.792 \pm 0.095$ | $0.917 \pm 0.071$ |
| GPT-4o-mini | $0.923 \pm 0.134$ | $0.718 \pm 0.045$ | $0.889 \pm 0.138$ | $0.778 \pm 0.100$ | $0.792 \pm 0.130$ | $0.875 \pm 0.110$ |
| O1 | $1.000 \pm 0.000$ | $1.000 \pm 0.000$ | $1.000 \pm 0.000$ | $1.000 \pm 0.000$ | $1.000 \pm 0.000$ | $0.833 \pm 0.071$ |
| O1-mini | $0.846 \pm 0.155$ | $0.744 \pm 0.221$ | $0.956 \pm 0.077$ | $0.911 \pm 0.077$ | $0.708 \pm 0.032$ | $0.813 \pm 0.063$ |
| Gemini 2.0 pro | $0.949 \pm 0.089$ | $1.000 \pm 0.000$ | $0.978 \pm 0.032$ | $1.000 \pm 0.000$ | $0.958 \pm 0.032$ | $1.000 \pm 0.000$ |
| Gemini 2.0 Flash | $0.974 \pm 0.045$ | $0.923 \pm 0.134$ | $0.933 \pm 0.063$ | $1.000 \pm 0.000$ | $0.854 \pm 0.145$ | $0.958 \pm 0.071$ |
| Gemini 2.5 Pro | $0.949 \pm 0.089$ | $1.000 \pm 0.000$ | $1.000 \pm 0.000$ | $1.000 \pm 0.000$ | $0.949 \pm 0.089$ | $1.000 \pm 0.000$ |
| **GED-nx (10s) ($\downarrow$)** | | | | | | |
| GPT-4o | $13.0 \pm 4.00$ | $10.67 \pm 1.53$ | $10.0 \pm 2.00$ | $8.67 \pm 4.04$ | $14.0 \pm 5.29$ | $16.33 \pm 4.04$ |
| GPT-4o-mini | $19.33 \pm 8.08$ | $20.67 \pm 5.77$ | $12.67 \pm 1.15$ | $16.00 \pm 3.61$ | $17.33 \pm 5.03$ | $15.33 \pm 5.13$ |
| O1 | $6.67 \pm 2.31$ | $5.33 \pm 1.15$ | $10.67 \pm 5.77$ | $8.00 \pm 3.46$ | $14.00 \pm 6.93$ | $13.33 \pm 1.15$ |
| O1-mini | $22.67 \pm 2.31$ | $12.00 \pm 5.57$ | $10.67 \pm 3.05$ | $13.00 \pm 1.73$ | $18.00 \pm 2.65$ | $18.00 \pm 0.00$ |
| Gemini 2.0 pro | $8.33 \pm 2.08$ | $7.00 \pm 1.41$ | $10.00 \pm 3.46$ | $6.67 \pm 2.31$ | $14.00 \pm 2.00$ | $12.67 \pm 1.15$ |
| Gemini 2.0 Flash | $16.33 \pm 8.02$ | $13.33 \pm 1.53$ | $11.00 \pm 1.00$ | $11.33 \pm 6.43$ | $17.00 \pm 6.08$ | $14.00 \pm 2.65$ |
| Gemini 2.5 Pro | $8.33 \pm 2.08$ | $5.00 \pm 1.73$ | $8.00 \pm 4.00$ | $3.33 \pm 2.31$ | $13.33 \pm 1.15$ | $9.50 \pm 2.12$ |
| **Spectral Distance ($\downarrow$)** | | | | | | |
| GPT-4o | $0.640 \pm 0.295$ | $0.772 \pm 0.447$ | $0.469 \pm 0.105$ | $0.383 \pm 0.286$ | $1.313 \pm 0.346$ | $0.946 \pm 0.622$ |
| GPT-4o-mini | $1.704 \pm 0.640$ | $1.482 \pm 0.462$ | $0.745 \pm 0.447$ | $1.072 \pm 0.362$ | $1.428 \pm 0.363$ | $1.055 \pm 0.510$ |
| O1 | $0.271 \pm 0.077$ | $0.271 \pm 0.077$ | $0.634 \pm 0.158$ | $0.567 \pm 0.451$ | $1.005 \pm 0.032$ | $1.261 \pm 0.265$ |
| O1-mini | $1.993 \pm 0.288$ | $1.219 \pm 0.564$ | $0.670 \pm 0.155$ | $0.636 \pm 0.412$ | $1.624 \pm 0.045$ | $1.576 \pm 0.071$ |
| Gemini 2.0 pro | $0.453 \pm 0.192$ | $0.431 \pm 0.190$ | $0.362 \pm 0.349$ | $0.192 \pm 0.000$ | $0.842 \pm 0.454$ | $1.064 \pm 0.071$ |
| Gemini 2.0 Flash | $1.026 \pm 1.123$ | $0.901 \pm 0.148$ | $0.501 \pm 0.197$ | $0.289 \pm 0.138$ | $1.137 \pm 0.497$ | $0.561 \pm 0.479$ |
| Gemini 2.5 Pro | $0.453 \pm 0.192$ | $0.310 \pm 0.029$ | $0.388 \pm 0.205$ | $0.199 \pm 0.032$ | $0.977 \pm 0.134$ | $1.052 \pm 0.055$ |

Table 5: AUROC Scores for Gene Regulatory Network Inference.

| LLM | Stomach | | Liver | | Kidney | |
|---|---|---|---|---|---|---|
| | **Direct** | **SCPILOT** | **Direct** | **SCPILOT** | **Direct** | **SCPILOT** |
| GPT-4o | $0.623 \pm 0.001$ | $0.800 \pm 0.001$ | $0.577 \pm 0.001$ | $0.743 \pm 0.000$ | $0.570 \pm 0.001$ | $0.707 \pm 0.002$ |
| GPT-4o-mini | $0.583 \pm 0.000$ | $0.697 \pm 0.001$ | $0.600 \pm 0.004$ | $0.683 \pm 0.001$ | $0.567 \pm 0.000$ | $0.733 \pm 0.001$ |
| O1 | $0.827 \pm 0.002$ | $0.873 \pm 0.004$ | $0.753 \pm 0.001$ | $0.760 \pm 0.000$ | $0.777 \pm 0.000$ | $0.797 \pm 0.001$ |
| O1-mini | $0.690 \pm 0.001$ | $0.783 \pm 0.002$ | $0.660 \pm 0.001$ | $0.700 \pm 0.000$ | $0.640 \pm 0.000$ | $0.727 \pm 0.001$ |
| Gemini 2.0 Pro | $0.949 \pm 0.008$ | $1.000 \pm 0.000$ | $0.600 \pm 0.000$ | $0.737 \pm 0.000$ | $0.600 \pm 0.000$ | $0.743 \pm 0.001$ |
| Gemini 2.0 Flash | $0.690 \pm 0.009$ | $0.697 \pm 0.003$ | $0.690 \pm 0.001$ | $0.753 \pm 0.003$ | $0.683 \pm 0.003$ | $0.730 \pm 0.004$ |
| Gemini 2.5 Pro | $0.610 \pm 0.006$ | $0.820 \pm 0.000$ | $0.637 \pm 0.000$ | $0.753 \pm 0.001$ | $0.623 \pm 0.001$ | $0.727 \pm 0.002$ |

three benchmark datasets. Among *direct* approaches, o1 achieved the highest overall accuracy (0.667 on PBMC3k, 0.560 on Liver, 0.474 on Retina), underscoring the importance of model capacity.

Implementing the SCPILOT's pipeline further improved accuracy for 19 out of 24 model–dataset combinations. The Retina dataset showed the most significant median accuracy gain (+0.180), followed by PBMC3k (+0.042) and Liver (+0.024). The sub-

Table 6: Trajectory metrics across methods

| Metric | SCPILOT Gemini-2.5 Pro | Biomni Gemini-2.5 Pro | py-Monocle |
|---|---|---|---|
| Jaccard ($\uparrow$) | 1 | 1 | 1 |
| GED-nx (10 s) ($\uparrow$) | $3.33 \pm 2.31$ | $8.33 \pm 3.21$ | 20 |
| Spectral Distance ($\downarrow$) | $0.199 \pm 0.033$ | $0.482 \pm 0.379$ | 0.469 |

stantial improvement in Retina is largely attributed to SCPILOT's iterative reasoning process, which effectively differentiated major cell populations such as *rod photoreceptors, Müller glia, and bipolar cells* by accessing the data and evaluating based on dotplot expression. Conversely, the one-step *direct* approach, limited to top marker genes, struggled with such detailed distinctions. Overall, SCPILOT

implementations using the `o1` and `Gemini-2.0-Pro` models ranked highest (0.792 on PBMC3k, 0.728/0.763 on Retina, 0.518/0.509 on Liver).

**Trajectory Inference.** Table 6 demonstrates SCPILOT's superior performance compared to baseline methods Biomni and Monocle. Comprehensive evaluation metrics—including node overlap (Jaccard), graph-edit distance (GED)-structure-aware scores, and spectral distance—are summarized in Table 4. For the *direct* approach, the `o1` model and `Gemini-2.5-Pro` achieved the best performance, with `o1` obtaining perfect Jaccard scores (1.000) and superior structural accuracy on Pancreas and Neocortex.

When adopting the SCPILOT pipeline, structural errors were further reduced in 10 of 21 model–metric pairs (median improvements: GED -2.0, spectral distance -0.14). `Gemini-2.5-Pro` consistently delivered optimal results, closely followed by `Gemini-2.0-Pro`.

**GRN TF-Gene Prediction.** Average AUROC results for GRN prediction across three tissues are summarized in Table 5. Table 7 provides baseline comparisons against two types of methods: graph neural network (GNN) models trained on GRN data and LLM-based tools (LLM4GRN, BioGPT). The SCPILOT pipeline consistently outperformed these baseline methods, except when utilizing smaller models (`GPT-4o-mini`, `Gemini-2.0-Flash`). Compared to *direct* prompting, SCPILOT demonstrated an average AUROC improvement of +0.098.

Again, the `o1` model under the SCPILOT pipeline achieved the highest overall accuracy (AUROC: 0.873 stomach, 0.760 liver, 0.797 kidney), with `Gemini-2.5-Pro` ranking second. `GPT-4o` exhibited the greatest relative improvement (+0.162 average AUROC), underscoring the effectiveness of iterative reasoning in harnessing latent regulatory insights.

Table 7: GRN prediction performance on Stomach dataset across methods. Note that BioGPT* refers to `BioGPT-Large-PubMedQA`

| Method | AUROC (Stomach) |
|---|---|
| GCN | 0.723 $_{\pm 0.071}$ |
| GraphSAGE | 0.713 $_{\pm 0.063}$ |
| GAT | 0.683 $_{\pm 0.071}$ |
| LLM4GRN | 0.727 $_{\pm 0.025}$ |
| BioGPT* | 0.660 |
| Direct (o1) | 0.827 $_{\pm 0.002}$ |
| SCPILOT (o1) | 0.873 $_{\pm 0.004}$ |

**Cross-Task Trends.** Three consistent patterns emerged across tasks: (1) Superior results arise from combining large-scale models (e.g., `o1`, `Gemini 2.0/2.5 Pro`) with structured, iterative reasoning in SCPILOT. (2) Mini or latency-optimized variants frequently produced unreliable outputs, including over-generation and hallucination, during extended reasoning chains. Overall, the results clearly indicate that iterative, reflective prompting significantly elevates state-of-the-art LLMs from competitive to decisively outperforming traditional bioinformatics methods in annotation, trajectory inference, and GRN prediction. (3) In rare cases, simpler *direct* prompting surpassed SCPILOT; these instances, though uncommon, offer valuable insights, discussed systematically in Appendix C.1.

**Challenges with Local Open-Source LLMs.** While open-source LLMs offer model transparency and data control advantages, our assessment of `Gemma-3` for automated annotation tasks highlighted critical limitations. Performance evaluations revealed consistent inferiority to proprietary models such as `GPT-4o` and `Gemini` (Table 3), suggesting that significant domain-specific fine-tuning is essential for accurate biological reasoning. Computational efficiency posed additional challenges: inference on the PBMC3k dataset required 135.7 seconds per evaluation using four NVIDIA A100 (80 GB) GPUs, compared to only 8.8 seconds for `GPT-4o`—a more than 15-fold difference. The combination of high hardware demand, prolonged runtime, and limited predictive accuracy renders fully on-premise deployments financially and operationally impractical for most laboratories. Thus, SCPILOT employs API-based models as its backbone, while local open-source LLMs were not pursued further.

**Efficiency and Cost Analysis**. To evaluate the practical accessibility of SCPILOT, we conducted a detailed cost and efficiency analysis using `Gemini-2.5-Pro`. As summarized in Table 13, executing the most complex tasks requires minimal financial outlay—mere cents per task—highlighting the affordability and scalability of our framework. Token counts were approximated using *tiktoken*, with cost rates of \$1.25 per million input tokens and \$10 per million output tokens, without caching. Furthermore, compared to the general-purpose agent Biomni, SCPILOT achieves up to **30× lower cost** and substantially faster performance due to its targeted reasoning and optimized toolchain (Table 20). Importantly, SCPILOT consistently succeeds in complex tasks like GRN prediction, where Biomni often fails, underscoring SCPILOT's advantage in specialized biological reasoning. Overall, our analyses demonstrate that SCPILOT offers an accessible and economically viable platform, significantly outperforming general-purpose methods in efficiency, cost, and reliability.

## 4.2 Ablation Studies

We performed three ablation experiments (Figure 3, Figure 4) rigorously assess the contributions of contextual metadata, domain context-Gene Ontology (GO) knowledge, and trajectory priors to the overall accuracy and robustness of SCPILOT 's reasoning.

**Contextual Metadata Ablation.** We first investigated the impact of dataset-level contextual metadata (such as dataset size, tissue origin, and experimental conditions) on cell-type annotation accuracy using the PBMC3k dataset. The full SCPILOT pipeline achieved strong baseline accuracy, with scores of 0.792 (o1), 0.646 (GPT-4o), and 0.604 (4o-mini). Removing contextual metadata led to noticeable declines in accuracy: 0.104 points (o1), 0.063 points (GPT-4o), and 0.188 points

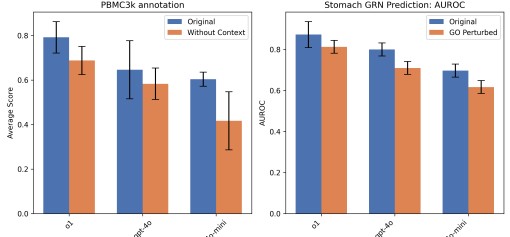

Figure 3: Ablation on metadata and GO context.

(4o-mini). Despite these performance drops, all models continued to outperform traditional single-cell baselines, indicating intrinsic robustness of LLM reasoning even with incomplete metadata. These findings underscore two critical observations: (i) high-capacity models such as o1 significantly depend on contextual information for precise biological interpretation, and (ii) smaller models, notably 4o-mini, exhibit heightened sensitivity to absent metadata. Thus, comprehensive contextual metadata integration is crucial, particularly for smaller or mid-sized LLMs, to facilitate biologically coherent reasoning.

**Gene Ontology Perturbation.** Next, we assessed the significance of accurate Gene Ontology (GO) information by perturbing the GO database utilized in the GRN prediction module. Specifically, genuine transcription-factor–gene annotations were randomized to simulate erroneous overlaps. This perturbation resulted in substantial reductions in GRN prediction accuracy: in the Stomach dataset, AUROC decreased from 0.873 to 0.813 (o1), from 0.800 to 0.710 (GPT-4o), and from 0.697 to 0.617 (GPT-4o-mini). Although accuracy declined, all models maintained higher performance compared to direct-prompting baselines, suggesting even perturbed GO data provides partial structural guidance. These results highlight that precise GO annotations between TFs and target genes are essential for robust GRN inference, with smaller models disproportionately affected by annotation inaccuracies.

**Trajectory Input Perturbation.** Finally, we evaluated the role of trajectory priors by corrupting the *py-Monocle* input used in the liver dataset's trajectory inference task. *Py-Monocle* typically provides statistical insights into intercluster relationships and pseudotime hierarchy, crucial for informing accurate trajectory reasoning. The corrupted Monocle inputs led to reduced reasoning accuracy for both o1 and

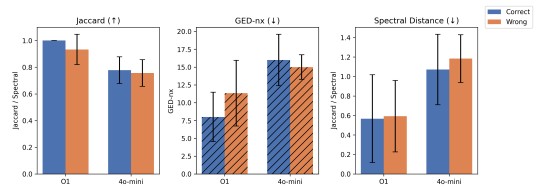

Figure 4: Ablation on trajectory inference input.

GPT-4o-mini based SCPILOT models relative to unperturbed conditions. These findings confirm the importance of accurate trajectory cues—specifically cluster connectivity and pseudotime structure—in enhancing LLM-based biological reasoning. Consequently, robust integration with established computational tools like *Monocle* is essential for high interpretability and analytical performance.

## 4.3 Biological Interpretability and Insights

We further investigated how SCPILOT transforms benchmark accuracy into biologically interpretable insights across representative single-cell analysis tasks using the SCBENCH evaluation suite.

**Multi-gene logic resolves marker ambiguity.** In cell-type annotation tasks, SCPILOT leverages an iterative *propose → filter → solve* reasoning loop, enabling systematic construction and validation of combinatorial marker hypotheses (Figure 5). Specifically, on the PBMC3k dataset, the o1-based SCPILOT model initially proposed candidate marker sets (e.g., NK cells: *NKG7*, *GNLY*, *GZMB*; CD8 T cells: *CD3D*, *CD8A*), filtered out absent markers (e.g., excluding plasma cells due to missing *SDC1*), and resolved marker expression ambiguity through dotplot reasoning. This meticulous process resulted in correct annotation for 7 out of 8 clusters. The model notably identified that *NKG7* alone is insufficiently specific; however, combining it with *CD3D* and *GNLY* reliably distinguishes NK cells from cytotoxic T cells—an important distinction often overlooked by single-marker methodologies,

particularly when *CD8A* exhibits weak expression. Further qualitative analysis and case studies are detailed in Appendix D.1.

**Self-auditing via monocle diagnostics sharpens trajectory accuracy.** For trajectory inference tasks, `Gemini-2.5-Pro`-based SCPILOT first constructed an initial lineage tree and subsequently self-audited the inferred structure utilizing *py-Monocle* diagnostic outputs. Through targeted refinements—including correction of the tree root, restoration of canonical hepatic lineage sequences, and hierarchical adjustments—SCPILOT substantially improved trajectory accuracy, reducing the GED-nx metric by six edits and decreasing the Spectral Distance by 0.32. These improvements resulted in a trajectory structure closely aligned with the developmental topology established in original biological studies. Comprehensive edge-level analyses are provided in Appendix D.2.

**Tissue-specific TF reasoning in GRN prediction.** In GRN prediction, SCPILOT effectively employed a tissue-specific retrieval module that filtered out spurious Gene Ontology (GO) overlaps, seamlessly integrating expression context with known regulatory pathway information. This approach demonstrated biologically informed, context-sensitive reasoning. For instance, in predicting *Stat1*'s regulation of *Irf7*, SCPILOT accurately captured the GO functional overlap. The prediction of *Klf4* regulating *Muc5ac* illustrated the model's nuanced tissue-specific understanding. More examples of successful and unsuccessful predictions are discussed thoroughly in Appendix D.3.

In summary, SCPILOT not only achieved competitive accuracy (e.g., a score of 0.789 on the retina atlas) but also elucidated the mechanistic bases underlying its predictions. By clearly identifying ontology gaps, marker ambiguities, and rare-cell misclassifications, SCPILOT delivers diagnostic transparency, significantly advancing biological interpretability. This transparency not only enhances biological discovery but also informs the iterative development of future omics-native reasoning models.

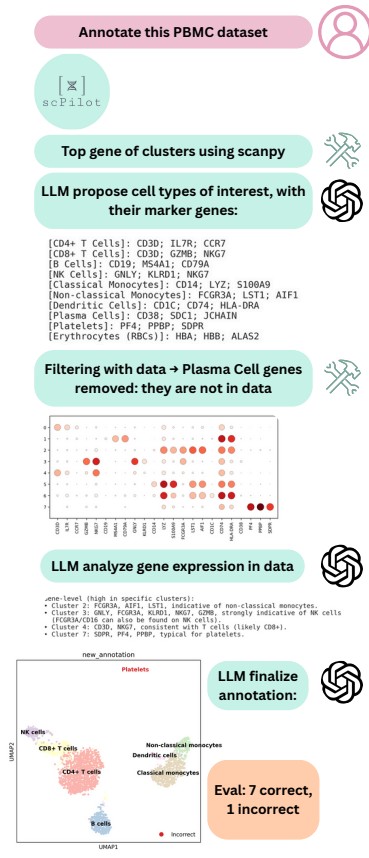

Figure 5: Example of SCPILOT multi-gene reasoning in PBMC3k annotation.

## 5 Conclusions

We introduced Omics-Native Reasoning (ONR), a novel LLM scientific reasoning paradigm wherein LLMs directly inspect raw single-cell data, invoke specialized analytic tools, and clearly articulate every biological inference in natural language. Operationalized through SCPILOT, ONR redefines critical analytical processes—cell-type annotation, trajectory inference, and gene regulatory network prediction—by transitioning them from opaque, black-box methods to transparent and interpretable conversational workflows. Our newly established benchmark, SCBENCH, quantitatively demonstrates significant improvements: iterative reasoning via SCPILOT employing the `o1` model boosts annotation accuracy by 11%, while using `Gemini-2.5-Pro` reduces trajectory graph-edit distance by 30% compared to traditional one-shot prompting approaches. By shifting analytical processes from implicit heuristics to explicit, data-informed reasoning, SCPILOT offers a robust, transparent, and continually improving foundation for single-cell biological discovery, paving the way for AI systems that actively reason alongside scientists as genuine scientific partners.

**Limitations and Future Work.** Despite progress, significant challenges persist. Current data compression may miss subtle signals from rare cell populations, requiring enhanced representation methods. Scaling ONR to billion-token contexts and integrating retrieval-augmented reasoning chains pose major scalability issues. Furthermore, ensuring trustworthiness requires robust methods to mitigate LLM hallucinations and incorrect claims. Finally, a key future direction is integrating experimental wet-lab feedback to validate computational predictions in vitro, confirming the biological validity of ONR frameworks.

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

# A  Data and Code Availability

All of the raw and processed single-cell RNA-seq datasets used in SCPILOT are publicly sourced collections. The Cell type annotation datasets are detailed in Table 9, the Trajectory inference collections in Table 10, and the GRN TF–gene prediction cohorts in Table 11. For each dataset, we release the processed objects (H5AD format for scRNA datasets, and csv format for GRN data), along with cell-type labels, trajectory gold-standards, and GRN reference networks.

The complete source code for dataset preprocessing, automatic graders, evaluation metrics, and benchmark drivers is released under the MIT license and available at our SCPILOT github `https://github.com/maitrix-org/scPilot`.

# B  Additional Details of SCPILOT and SCBENCH

## B.1  Model zoo

We specify the exact versions of the 7 proprietary models and 1 open-source model in the SCPILOT. This model zoo spans both large-scale multimodal systems and lightweight, inference-optimized variants, ensuring a comprehensive evaluation of performance trade-offs in single-cell analysis tasks.

Table 8: Large Language Model Zoo: Current State-of-the-Art Models

| Model Name | Version/ID | Description |
|---|---|---|
| *OpenAI Models* | | |
| GPT-4o | `gpt-4o-2024-08-06` | Omni model with multimodal capabilities |
| GPT-4o-mini | `gpt-4o-mini-2024-07-18` | Lightweight variant of GPT-4o |
| O1 | `o1-2024-12-17` | Advanced reasoning model |
| O1-mini | `o1-mini-2024-09-12` | Efficient version of O1 |
| *Google DeepMind Models* | | |
| Gemini 2.0 Pro | `gemini-2.0-pro-exp-02-05` | Experimental professional model |
| Gemini 2.0 Flash | `gemini-2.0-flash-thinking-exp-01-21` | Optimized for fast inference |
| Gemini 2.5 Pro | `gemini-2.5-pro-exp-03-25` | Latest pro model iteration |
| *Google Open Models* | | |
| Gemma 3 27B | — | 27B parameter instruction-tuned model |

## B.2  Dataset description

We elaborate on the details of the dataset in SCBENCH.

**Cell Type Annotation**. We selected 3 datasets, and their details are in 9. The "# Cell types" column refers to the number of ground truth cell types. The "Celltypist Model" column refers to the Celltypist model we used to evaluate Celltypist baseline performance on this dataset.

Table 9: Summary of Cell Type Annotation Datasets

| Dataset | Tissue | Size | # Cell Types | CellTypist Model | Source |
|---|---|---|---|---|---|
| Liver | Mouse liver | 41,000 cells × 2,000 genes (HVGs) | 31 | `Healthy_Mouse_Liver` | [43] |
| PBMC3k | Human PBMC | 2,638 cells × 13,714 genes | 8 | `Immune_All_Low` | [1] |
| Retina | Human retina | 20,091 cells × 19,719 genes | 9 | `Fetal_Human_Retina` | [50] |

**Trajectory Inference**. We selected 3 datasets, and their details are in 10. The "# Timepoints" column represents the innate developmental stage design in the single-cell sequencing experiment. For example, in the Pancreas dataset, there are 4 timepoints, from E12.5 to E15.5, corresponding to Day 12.5 to Day 15.5 in the embryo. The "# Trajectory Nodes" represents the number of cell type, and thus the number of nodes on the trajectory tree.

**GRN TF-gene Prediction**. From the GRNdb database, we selected 3 tissues, and their details are in 11. For each tissue, we compared its SCENIC-generated TF-gene pairs with the CHiP-Seq verified

Table 10: Summary of Trajectory Datasets

| Dataset | Tissue | Size | # Timepoints | # Trajectory Nodes | Source |
|---|---|---|---|---|---|
| Pancreas | Mouse pancreas | 36,351 cells × 17,327 genes | 4 | 14 | [5] |
| Liver | Mouse liver | 44,010 cells × 2,000 genes (HVGs) | 4 | 15 | [45] |
| Neocortex | Human neocortex | 33,976 cells × 35,543 genes | 2 | 16 | [53] |

database TRRUST. If this pair exists in TRRUST, we recognize it as a positive edge. Then, we randomly sample another gene that does not result in a verified or SCENIC-generated TF-gene edge. So we have half questions as positive (answer is yes, TF-gene relationship exists) and half as negative (answer is no, there is no regulation).

## B.3 Evaluation Metrics

### B.3.1 Cell Type Annotation

The key to calculating the Cluster-level accuracy is similarity based on the GO database.

**Cleaning and Standardizing Cell Type Names**. Raw cell type names often exhibit inconsistencies such as mixed casing, redundant suffixes, or ambiguous abbreviations. To address this, a cleaning and standardization process was applied:

Table 11: GRN task summary by held-out context

| Dataset name | Verified TF–gene Edge | Total Questions | Unique TFs |
|---|---|---|---|
| Stomach | 23 | 46 | 10 |
| Liver | 71 | 142 | 21 |
| Kidney | 49 | 98 | 21 |

- **Automatic Cleaning:** Plural forms (e.g., "cells") were converted to singular (e.g., "cell"), redundant whitespace and punctuation were removed, and biologically meaningful symbols (e.g., slashes "/") were retained.

- **Standardized Mapping:** Cleaned names were first matched against a predefined dictionary of known cell type nomenclature. For names not found in the dictionary, a language model was used to generate standardized mappings dynamically, ensuring coverage of uncommon or ambiguous terms.

This process harmonized all cell type names, enabling consistent downstream analysis.

**Mapping to Cell Ontology Identifiers**. Standardized cell type names were mapped to terms in the Cell Ontology (CL) to integrate annotations with structured biological knowledge:

- **Ontology Querying:** Names were queried against the Cell Ontology using an automated search tool, retrieving corresponding identifiers (CLIDs) and high-level categories.

- **Handling Unmapped Names:** Names that could not be mapped directly to the ontology were retained for further review or additional processing.

This mapping ensured systematic alignment between predicted and reference annotations.

**Construction of the Ontology Tree**. The hierarchical relationships within the Cell Ontology were leveraged to evaluate lineage-based relationships between predicted and reference annotations:

- **Ontology Hierarchy Parsing:** The Cell Ontology was downloaded in OWL format, and parent-child relationships between ontology terms were extracted.

- **Graph Construction:** A directed acyclic graph (DAG) was built, where nodes represented CLIDs, and edges denoted parent-child relationships.

This hierarchical structure enabled evaluation beyond direct matches, incorporating extended lineage-based relationships.

**Ontology-Based Scoring Framework**. Inspired by the ontology-based scoring methodology in GPTcelltype, a scoring framework was developed to incorporate both exact and hierarchical matches:

- **Exact Match:** A score of 1.0 was assigned if the predicted CLID(s) exactly matched the reference CLID(s).

- **Partial Match:** A score of 0.5 was assigned if the predicted CLID(s) overlapped with the reference CLID(s) or their relatives (parent or child terms) in the ontology hierarchy.

- **No Match:** A score of 0.0 was assigned if no overlap was observed.

This scoring framework ensured biologically meaningful comparisons while accounting for the hierarchical structure of the Cell Ontology.

**Biological Context Validation**. To ensure the biological relevance of predictions, an additional layer of validation was performed:

- **Relative Identification:** Using the ontology graph, predictions were cross-checked against all known relatives of the reference CLIDs, including parents and children.

- **Broad Type Consistency:** Predictions were compared at higher categorization levels (e.g., "immune cells," "stromal cells") to ensure consistency with the broader biological context.

**Summary**. The evaluation methodology integrates systematic name cleaning, dynamic mapping, and ontology-aware scoring to ensure biologically accurate assessments. By leveraging both direct matches and hierarchical relationships within the Cell Ontology, the framework provides a robust and biologically meaningful evaluation of cell type annotations.

### B.3.2 Trajectory Inference

To evaluate the predicted trajectory tree against the ground truth, we employ three distinct metrics capturing structural and spectral similarities:

**Jaccard Similarity (Nodes)**. We quantify the structural similarity between predicted and ground truth trees at the node level using the Jaccard similarity coefficient. For two trajectory graphs with node sets $V_{\text{pred}}$ and $V_{\text{gt}}$, representing the predicted and ground truth trees, respectively, the Jaccard similarity is defined as:

$$J(V_{\text{pred}}, V_{\text{gt}}) = \frac{|V_{\text{pred}} \cap V_{\text{gt}}|}{|V_{\text{pred}} \cup V_{\text{gt}}|} \tag{7}$$

where $|\cdot|$ denotes set cardinality. This metric ranges from 0 to 1, with higher values indicating greater overlap between the node sets of the two trees.

**Graph Edit Distance (GED)**. Graph Edit Distance (GED) quantifies the structural dissimilarity between two graphs by measuring the minimum number of edit operations required to transform one graph into another. For predicted and ground truth graphs $G_{\text{pred}} = (V_{\text{pred}}, E_{\text{pred}})$ and $G_{\text{gt}} = (V_{\text{gt}}, E_{\text{gt}})$, the GED is formally defined as:

$$\text{GED}(G_{\text{pred}}, G_{\text{gt}}) = \min_{\gamma \in \Gamma(G_{\text{pred}}, G_{\text{gt}})} \sum_{e \in \gamma} c(e) \tag{8}$$

where $\Gamma(G_{\text{pred}}, G_{\text{gt}})$ denotes the set of all possible edit paths transforming $G_{\text{pred}}$ into $G_{\text{gt}}$, and $c(e)$ represents the cost of edit operation $e$ (node/edge insertion, deletion, or substitution). For computational tractability, we impose a timeout constraint of 10 seconds. Lower GED values indicate greater structural similarity between the graphs.

**Spectral Distance (Euclidean)**. Spectral distance captures global structural properties of graphs by comparing the eigenvalue distributions of their normalized Laplacian matrices. For graphs $G_{\text{pred}}$ and $G_{\text{gt}}$, let $\mathcal{L}_{\text{pred}}$ and $\mathcal{L}_{\text{gt}}$ denote their respective normalized Laplacian matrices. The spectral distance is defined as the Euclidean distance between their ordered eigenvalue spectra:

$$d_{\text{spectral}}(G_{\text{pred}}, G_{\text{gt}}) = \|\boldsymbol{\lambda}_{\text{pred}} - \boldsymbol{\lambda}_{\text{gt}}\|_2 = \sqrt{\sum_{i=1}^{n} (\lambda_i^{\text{pred}} - \lambda_i^{\text{gt}})^2} \tag{9}$$

where $\boldsymbol{\lambda}_{\text{pred}} = (\lambda_1^{\text{pred}}, \dots, \lambda_n^{\text{pred}})$ and $\boldsymbol{\lambda}_{\text{gt}} = (\lambda_1^{\text{gt}}, \dots, \lambda_n^{\text{gt}})$ are the eigenvalues of $\mathcal{L}_{\text{pred}}$ and $\mathcal{L}_{\text{gt}}$ arranged in ascending order. Lower spectral distances indicate greater similarity in the global topological structure of the graphs.

### B.3.3 GRN TF-gene Prediction

We evaluate the prediction performance of Gene Regulatory Network (GRN) transcription factor (TF)-gene interactions using standard binary classification metrics:

**Area Under the ROC Curve (AUROC).** The AUROC quantifies the classifier's discriminative ability across all possible decision thresholds. For a binary classifier with varying threshold $\tau$, the AUROC is computed as:

$$\text{AUROC} = \int_0^1 \text{TPR}(t)\, d\text{FPR}(t) \tag{10}$$

where the True Positive Rate (sensitivity) and False Positive Rate (1-specificity) are defined as:

$$\text{TPR}(\tau) = \frac{\text{TP}(\tau)}{\text{TP}(\tau) + \text{FN}(\tau)}, \quad \text{FPR}(\tau) = \frac{\text{FP}(\tau)}{\text{FP}(\tau) + \text{TN}(\tau)} \tag{11}$$

**Confusion Matrix.** The confusion matrix provides a comprehensive breakdown of prediction outcomes for a given threshold, capturing the counts of true negatives (TN), false positives (FP), false negatives (FN), and true positives (TP):

$$\mathbf{C} = \begin{bmatrix} \text{TN} & \text{FP} \\ \text{FN} & \text{TP} \end{bmatrix} = \begin{bmatrix} |\hat{y}=0, y=0| & |\hat{y}=1, y=0| \\ |\hat{y}=0, y=1| & |\hat{y}=1, y=1| \end{bmatrix} \tag{12}$$

where $y$ and $\hat{y}$ denote the ground truth and predicted labels, respectively.

## C Additional Results

### C.1 Occasional suboptimal performance

Occasionally, a simpler 'Direct' prompt can outperform SCPILOT. These cases are rare, systematic, and informative. SCPILOT overwhelmingly outperforms the baseline (wins in 87 of 108 total comparisons). The rare losses are systematic: 13 of 21 (62%) are from less-capable "mini" models. Their limited capacity for sustained logic leads them to "over-explore" and make mistakes with SCPILOT's extended reasoning.

For the few remaining cases with powerful models, the cause is high dataset nuances, inducing "overthinking." The only two powerful-model losses in cell-type annotation occurred on our most complex dataset, Liver. The characteristics of our datasets explain this pattern:

Table 12: Runtime of various LLMs.

| LLM | Runtime | Std. Dev. |
|---|---|---|
| GPT-4o | 3.9701 | 0.1045 |
| GPT-4o-mini | 3.7513 | 0.1673 |
| o1 | 11.4210 | 0.4140 |
| o1-mini | 4.0072 | 0.1827 |
| Gemini-2.0-pro | 30.6115 | 0.4351 |
| Gemini-2.0-flash | 11.0586 | 0.1445 |
| Gemini-2.5-pro | 29.7689 | 0.2197 |

**PBMC3k** contains 8 clusters and 8 cell types. It is a simple dataset characterized by a clear 1:1 mapping between clusters and cell types, with distinct marker genes. SCPILOT performs well on this dataset, although mini models may already saturate its performance ceiling.

**Retina** consists of 18 clusters and 9 cell types. It has moderately clear cell types with slight ambiguity. On this dataset, SCPILOT consistently improves accuracy over simpler baselines.

**Liver** includes 28 clusters and 31 cell types. This dataset is highly complex, with overlapping developmental lineages and noisy expression patterns. On such data, SCPILOT can sometimes "overthink" and merge distinct subtypes, reducing performance.

In the complex Liver dataset, SCPILOT using the powerful o1 model achieved a score of 0.518, while the simpler Direct method scored 0.560. A key error involved confusing developmentally related *hepatocytes* and *hepatoblasts*. Because they share many markers, SCPILOT's deep reasoning amplified this ambiguity and wrongly merged them, whereas the Direct method was incidentally correct by ignoring this nuance.

This analysis delineates the boundaries for applying LLMs to complex biological data and will be incorporated into the manuscript. Future improvements include adaptive reasoning depth and marker clarity assessments.

## C.2 Time Cost Analysis

In Table 12, we recorded average time cost of different LLM API calls, during the TF-gene prediction task. The OpenAI family consistently returns results in 3.8 - 4.0s except `o1`, whereas the full-scale `Gemini-2.0-pro` and `Gemini-2.5-pro` take about 30s per run—way slower than OpenAI models. Even Gemini "flash-thinking" ($\approx$11s) is still nearly three times slower than OpenAI's baseline models. `o1` reasoning took 3x extra time compared with its mini variant.

This significant runtime gap indicated clear trade-offs among reasoning depth, inference latency, and model scale. Gemini's substantial delay likely arose from Google AI's more complex internal processing or longer input handling, as even its optimized "flash-thinking" variant failed to match OpenAI's baseline speed. Meanwhile, `o1`'s roughly threefold increase in latency compared to its mini variant suggested that detailed step-by-step reasoning incurs notable overhead, potentially due to longer context handling and more elaborate token-by-token generation. Thus, selecting appropriate LLMs for biological tasks involved balancing performance demands: faster models like `GPT-4o` or mini variants for efficiency, versus slower, larger models when sophisticated reasoning outweighs runtime considerations. While Gemini was substantially slower in its AI infrastructure.

## C.3 Accessibility and Financial Cost Analysis

Our novel omics-native reasoning (ONR) paradigm is inherently more demanding than standard text analysis and requires powerful LLMs, a point confirmed by our experiments on `Gemma 3 27B` (the best open-source model testable on 8xH100s at the time). Our experiments reveal two reasons why ONR is challenging for current open-source models:

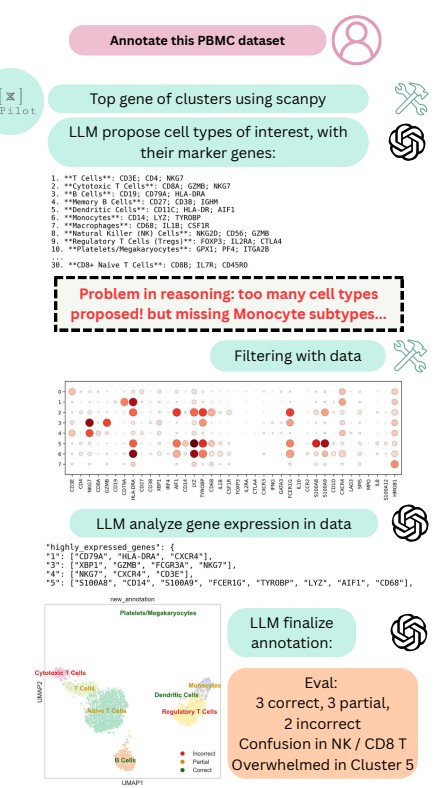

Figure 6: SCPILOT `GPT-4o-mini` reasoning in PBMC3k annotation, with problem noted

**Insufficient Domain Knowledge:** Weaker models lack the nuanced, pre-trained understanding of biology, such as raw markers and pathways, to interpret omics data correctly.

**Poor Instruction Following:** Weaker models often fail to follow complex instructions and adhere to biologist-defined formats (e.g., JSON), breaking the reasoning chain.

These insights are a key contribution, offering guidance for future model development. However, the cost of using powerful models via SCPILOT is negligible. Our detailed cost analysis for `Gemini-2.5-Pro` shows that a complete run for our most complex tasks costs only a few cents, making the framework accessible 13. Token counts were approximated with *tiktoken*. Cost for `Gemini-2.5-Pro` is 1.25 / 1M input, 10 / 1M output; no caching was used in the cost analysis. The negligible cost allows any lab to leverage state-of-the-art AI without expensive local GPUs. Developing cheaper, specialized open-source models for ONR is a promising future direction; in parallel, our framework provides an immediate, accessible tool for the community.

## C.4 Additional ablation studies

### C.4.1 Sensitivity analysis on Cell type annotation marker gene selection

For cell-type annotation, a key hyperparameter is $K$, the number of top marker genes. We tested performance sensitivity on the PBMC3k dataset for $K = 5$, 10 (our default), and 20 in table 14.

| Task (single run) | In token | Out token | In $ | Out $ | Total |
|---|---|---|---|---|---|
| Cell-type annotation (retina) | 6,155 | 2,197 | 0.008 | 0.022 | **0.03** |
| Trajectory inference (neocortex) | 8,221 | 2,702 | 0.010 | 0.027 | **0.04** |
| GRN TF→gene (stomach, 46 pairs) | 17,877 | 9,276 | 0.022 | 0.093 | **0.12** |

Table 13: Cost analysis of SCPILOT on all three tasks (`Gemini-2.5-Pro`)

Performance peaks at our chosen $K = 10$. More importantly, the strong results across different values demonstrate that our approach is robust and not highly sensitive to this hyperparameter.

Table 14: Annotation Accuracy (PBMC3k) vs. Number of Top Genes ($K$)

| $K$ | Model | Avg Accuracy | Variance |
|---|---|---|---|
| 5 | o1 | 0.771 | 0.001 |
| 5 | GPT-4o-mini | 0.500 | 0.016 |
| **10 (Default)** | o1 | **0.792** | **0.005** |
| **10 (Default)** | GPT-4o-mini | **0.604** | **0.001** |
| 20 | o1 | 0.750 | 0.016 |
| 20 | GPT-4o-mini | 0.542 | 0.009 |

### C.4.2 Validation of Intermediate Tool Calls

To validate that SCPILOT's reasoning is grounded in its tool outputs, we performed perturbation studies on two critical components. We will add these results to the manuscript.

**1. Perturbation of Gene Ontology (GO) Database in GRN Prediction** To test dependency on the GO database for Gene Regulatory Network (GRN) prediction, we shuffled its term associations with random noise 15. Corrupting GO data significantly degrades AUROC (e.g., $p = 0.044$ for `GPT-4o-mini`), confirming that SCPILOT's reasoning relies on accurate information from this intermediate step.

Table 15: Impact of GO Perturbation on GRN Prediction (AUROC)

| Model | AUROC (Original) | AUROC (GO Shuffled) | $\Delta$ AUROC |
|---|---|---|---|
| o1 | **0.873** | 0.813 | -0.060 |
| gpt-4o | **0.800** | 0.710 | -0.090 |
| GPT-4o-mini | **0.697** | 0.617 | -0.080 |

**2. Perturbation of py-Monocle Output in Trajectory Inference** We tested dependency on `py-Monocle` by providing SCPILOT with a corrupted report (randomized cluster relationships and pseudotime order) for the liver dataset 16. The performance drop across metrics demonstrates that accurate outputs from tools like Monocle are critical for SCPILOT.

**Summary:** These experiments prove that SCPILOT's success is fundamentally dependent on the integrity of the data provided by the bioinformatics tools it integrates.

### C.4.3 Additional Experiments in Error Propagation and Uncertainty Assessment

**1. Error Propagation via Input Perturbation** We highlight the *input perturbation experiments* from our main text (Figure 4a), where we removed all contextual metadata (e.g., tissue type) from the input prompt for the PBMC3k annotation task. In the Supplementary experiment, we removed the key input context, witnessing significant performance drops, and confirming that rich metadata is critical for reliability 17.

**2. Uncertainty and Reliability Assessment** We performed a new reliability analysis on the GRN prediction task (Stomach dataset). We calculated 95% confidence intervals (CIs) from 10 trials, performed 1000-bootstrap resampling, and assessed calibration using Expected Calibration Error

Table 16: Impact of Monocle Perturbation on Trajectory Inference (`o1` model)

| Metric | Original Avg | Perturbed Avg | Performance Change |
|---|---|---|---|
| Jaccard | **1.000** | 0.933 | Worse |
| GED-nx (10s) | **8.00** | 11.33 | Worse |
| Spectral Distance | **0.567** | 0.593 | Worse |

Table 17: Results of Context Removal on PBMC Dataset Annotation Accuracy

| Model | Accuracy (Original) | Accuracy (No Context) | Performance Drop |
|---|---|---|---|
| `o1` | **0.792** | 0.688 | ↓ **0.104** |
| GPT-4o | **0.646** | 0.583 | ↓ **0.063** |
| `GPT-4o-mini` | **0.604** | 0.416 | ↓ **0.188** |

(ECE) and Brier scores. The `o1` model is more accurate, stable (tighter CI), and reliable (lower ECE and Brier scores) 18.

Table 18: Uncertainty & Calibration Metrics for GRN Prediction

| Model | 95% CI (10 runs) | Bootstrap ($\pm$) | ECE | Brier Score |
|---|---|---|---|---|
| `o1` | **0.84–0.90** | $\pm 0.05$ | 0.05 | 0.14 |
| `GPT-4o-mini` | 0.65–0.75 | $\pm 0.10$ | 0.12 | 0.20 |

## C.5 SCPILOT vs. A General-purpose Biomedical Agent Biomni

We conducted a detailed qualitative and quantitative comparison against **Biomni**, a powerful state-of-the-art general-purpose biomedical LLM agent. This comparison highlights the value of SCPILOT's specialized, reasoning-first approach.

**Contrasting Design: Reasoning-First vs. Tool-First** The core difference is our design philosophy. SCPILOT is a *specialized, reasoning-first agent*, whereas Biomni is a *general-purpose, tool-first agent*. We described in detail in table 19.

**Impact on Scientific Accuracy** This design difference leads to significant accuracy gaps. In all 3 tasks, SCPILOT significantly outperformed Biomni. The results have been included in the main text tables 2, 6 and 7.

For two specific reasoning differences: In **Retina annotation:** SCPILOT correctly distinguishes fine-grained subtypes (e.g., ON- vs. OFF-bipolar cells), whereas Biomni makes clear errors, mislabeling Müller glia as amacrine cells. In **Liver trajectory inference:** SCPILOT correctly identifies the Epiblast root and reconstructs faithful lineages, while Biomni misplaces cell types (e.g., "Cardiac muscle") in the lineage, ignoring gene evidence.

**Value in Efficiency and Cost** SCPILOT's specialized approach is also vastly more efficient and cost-effective. As shown in table 20, SCPILOT is up to **30× cheaper** and

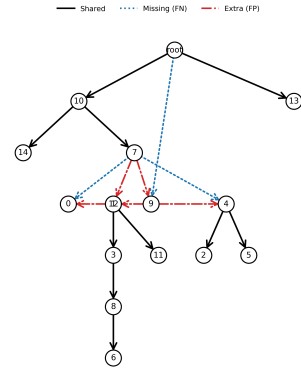

Figure 7: SCPILOT `o1` Liver Trajectory

significantly faster because its curated toolchain and reasoning-first method avoid costly, broad tool searches. It also succeeds on the complex Gene Regulatory Network (GRN) prediction task, where the general-purpose agent fails. This comparison shows that while general-purpose agents are flexible, SCPILOT's specialized reasoning delivers higher scientific fidelity at a fraction of the cost.

Table 19: Design comparison of SCPILOT (reasoning-first) and Biomni (tool-first).

| Dimension | SCPILOT (Reasoning-First) | Biomni (Tool-First) |
|---|---|---|
| **Workflow** | Hypothesis formulation → Omics-native reasoning → Iterative refinement | Run tool → Keyword lookup → Hard-coded template |
| **Error Handling** | Self-diagnoses biologically implausible steps and revises | Falls back to heuristics on Python errors; no biological validation |
| **Output** | **Transparent Chain-of-Thought:** Explanations, confidence scores | **Data Dump:** Dictionaries, tables, templated outputs |

Table 20: Efficiency and cost comparison between SCPILOT and Biomni.

| Task (vs. Biomni on Gemini-2.5 Pro) | SCPILOT Time | SCPILOT Cost | Biomni Cost | Cost Ratio |
|---|---|---|---|---|
| Cell-type annotation | 1–3 min | $0.03 | $0.80–$1.00 | $\sim$27–33$\times$ |
| Trajectory inference | 1–3 min | $0.04 | $0.80–$1.00 | $\sim$20–25$\times$ |
| GRN prediction | 3–5 min | $0.12 | Unsuccessful | N/A |

# D  Biological Insights

## D.1  Annotation

Figure 8 analyzes the component impact on retina annotation. SCPILOT GPT-4o scored 0.789 on retina, correctly annotating 14 clusters (e.g., Rod Photoreceptors), with 2 partially matched and 3 incorrect (e.g., Cluster 19). In cluster 0,1,7,8, the model correctly picked up rod-specific phototransduction cascade genes. Then, in cluster 14, the model demonstrated its ability to resolve low-abundance cones, despite a much larger rod pool. In cluster 17, SCPILOT successfully found horizontal Cells with expressions of genes like PROX1 and GAD1, meaning it was capable of correctly identifying rare cell types ($\approx$ less than 1% of retina cells).

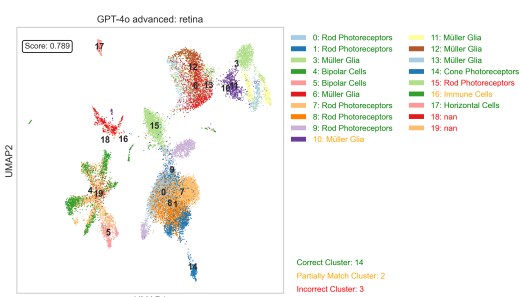

Figure 8: SCPILOT GPT-4o Retina Annotation

In the main text, we displayed the superior annotation reasoning of SCPILOT in SCPILOT using the o1 model. Here we present the inferior GPT-4o-mini reasoning proposed too many T cell subtypes and confused by multiple gene expressions 6: labeling *NKG7/GZMB/FCGR3A* + with CD8 T (Cytotoxic T Cells), and *NKG7*, *CXCR4*, *CD3E* with a broad T cell. In cluster 5, GPT-4o-mini collected 8 distinct highly expressed genes, and failed to find their relationship, labeling it with T cell again incorrectly.

## D.2  Trajectory Inference

Figure 7 shows a sample trajectory tree analysis in a liver dataset with SCPILOT utilizing the o1 model. It compares the prediction to the ground truth, with shared edges annotated in solid black. The sharing 10 edges, with 3 missing (false negatives, blue) and 5 extra (false positives, red), showing SCPILOT's role in structural accuracy. In Table 21, these ten faithful edges preserve the global skeleton of liver organogenesis: extra-embryonic tissues split early, mesoderm feeds into hepatic and vasculogenic branches, and the hepatoblast lineage matures correctly down to terminal hepatoblasts.

In Table 22, collectively, three missed and five extra edges all cluster around the mid-level mesodermal hub (nodes 7, 9, 12, 0, 4). The predictor correctly recognized the existence of these cell types (so Jaccard = 1) but mis-ordered their emergence, compressing or swapping developmental stages.

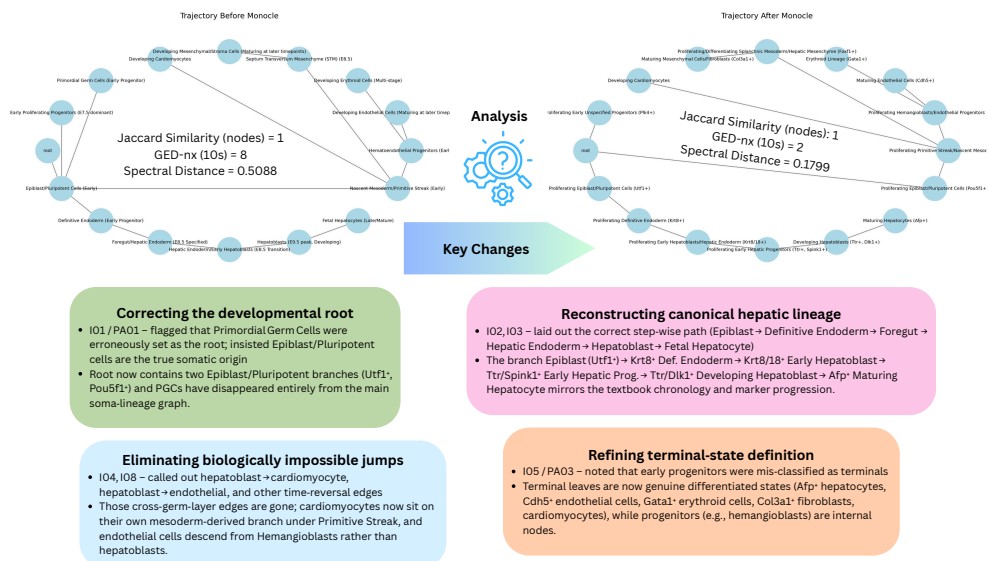

Figure 9: `Gemini-2.5-Pro` omics-driven reasoning in Trajectory Inference, with/without py-Monocle

Table 21: Strengths of the prediction

| Developmental branch | Correct edge(s) | Biological interpretation |
|---|---|---|
| Early extra-embryonic lineage | root→13 (visceral endoderm) | Visceral-endoderm branch is captured exactly. |
| Primitive streak II cascade | root→10→14, 10→7 (primitive streak II → yolk sac/mesoderm) | Temporal progression from primitive streak II into yolk-sac lineage and generic mesoderm reproduced. |
| Hepatic trajectory | 12→3→8→6, 12→11 | splanchnic mesoderm→ gut-tube endoderm → migrating→ definitive hepatoblasts; side branch to septum transversum mesenchyme Full liver lineage—including side mesenchymal branch—recovered without error. |
| Hemangioblast bifurcation | 4→2, 4→5 (endothelium / hematopoietic) | Correct divergence of endothelial vs. hematopoietic fates from hemangioblast. |

Concluding the biological picture in this predicted tree: Early specification (root → primitive streak II / visceral endoderm) and late hepatic maturation (hepatoblast lineage) were handled very well. Mesodermal diversification, the branching that creates cardiac, vasculogenic, and early streak I derivatives, was the main failure point, leading to an over-connected mesh among mesodermal descendants. Outside this mid-zone, no errors occurred. Thus, sCPILOT's largest contribution to the errors (3 FN + 5 FP) was a localized mis-routing rather than global disorganization.

In short, the predictor gave an accurate outline of liver development but compresses the timeline of mesodermal commitment, underscoring the need for finer temporal cues to distinguish closely related mesoderm-derived cell types. sCPILOT successfully integrated functional relevance, expression context, and known regulatory pathways to make accurate predictions, demonstrating their potential for biologically informed inference when provided with sufficient cues.

### D.3 GRN Prediction

The correct predictions in Table 23 were explained in the main text.

Table 22: Weak spots in the prediction

| Issue | Edge(s) concerned | Error type | Consequence |
|---|---|---|---|
| Primitive streak I orphaned | `root`→9 (primitive streak I) | Missing (FN) | Early streak I appears later than it should, breaking its parallel origin with streak II. |
| Mesodermal diversification under-represented | 7→0, 7→4 (cardiac mesoderm, hemangioblast) | Missing (FN) | Fails to branch cardiac and vasculogenic fates directly from mesoderm. |
| Spurious fan from mesoderm | 7→12, 7→9 | Extra (FP) | Mesoderm now feeds splanchnic mesoderm and streak I, conflating separate temporal stages. |
| Mis-routing via splanchnic mesoderm | 12→0, 12→9 | Extra (FP) | Pulls cardiac mesoderm and streak I into a downstream compartment. |
| Looping of vasculogenic branch | 9→4 (hemangioblast) | Extra (FP) | Places hemangioblast downstream of streak I instead of mesoderm, forming an artificial detour. |

The incorrect predictions 24 frequently arose from insufficient search for domain knowledge or misleading biological signals. For instance, the prediction of *Usf2* regulating *Pigr* occurred when models lacked detailed searching for their relationship, limiting their ability to justify or reject a link. In predicting *Fos* regulating *Hmox1*, shared GO term annotations between a TF and its candidate target gene falsely suggested functional association, leading to incorrect reasoning.

### D.4 Conclusion on SCPILOT accelerating discovery.

1. *Transparent agent loops.* All intermediate outputs and reasoning steps are logged, letting biologists audit each decision.
2. *Cross-task generality.* The same kernel framework solved annotation, trajectory inference, and GRN prediction tasks, hinting at broad utility.
3. *Plug-and-play tool integration.* The agent can ingest outputs from Monocle, SCENIC, and more traditional bioinformatics analysis tools without code changes, making it easy to layer domain heuristics on top of LLM reasoning.

## E Prompt Templates

### E.1 Cell Type Annotation Direct Prompting

Below is the one-step prompting for cell type annotation, purely based on the top marker genes for each cluster, and the context.

---
**Direct Prompt for Cell type Annotation**

system role = "You are expert in scRNA sequencing cell type annotation."
content = this is background information: self.config["initial_hypothesis"]
look at this dict: cluster_gene_dict. This is cluster number and the corresponding top differential genes of each cluster. Please provide cell type annotation for each cluster.
Output in text dict format just like the input dict. Keys are number of cluster, and Values are strings of cell type names. Output should be text dict, no other word should exist.

---

### E.2 Cell Type Annotation SCPILOT

Below are the prompts in the multi-agent framework for SCPILOT cell type annotation. In the hypothesis generation, SCPILOT integrates the top marker genes per cluster, dataset context and potentially, information from previous iterations to create hypothesis about the dataset.

Table 23: SCPILOT o1 successful reasoning in TF-Gene Prediction

| TF → Gene (Tissue) | Reasoning | Why the reasoning succeeded |
|---|---|---|
| Stat1 → Irf7 (Stomach) | Stat1 is a key regulator of interferon signaling, and Irf7 is essential for type I interferon induction. Both genes share the GO term defense response to virus, suggesting functional overlap. Although direct evidence in stomach tissue is limited, shared antiviral pathways make direct regulation plausible. | The answer recognizes that STAT1 sits at the hub of type-I interferon signaling and that IRF7 is the master amplifier of the same pathway. It explicitly points out their shared GO term "defense response to virus", connecting transcription-factor activation, downstream gene induction, and antiviral state. That functional convergence is exactly what experimental immunologists rely on when mapping STAT1–IRF7 regulatory cascades in mucosal tissues. |
| Klf4 → Muc5ac (Stomach) | Klf4 is known to maintain epithelial homeostasis and can regulate mucin genes in the gastrointestinal tract. Muc5ac is the principal mucin in the stomach and is crucial for gastric barrier function. Given Klf4's role in controlling mucin expression (e.g., Muc2) and epithelial differentiation, it is plausible that it may also directly regulate Muc5ac. However, definitive experimental evidence for direct regulation in the stomach remains limited. | Instead of merely noting that both are "epithelial," the model explains that KLF4 maintains gastric epithelial homeostasis and has precedent for activating mucin genes (e.g., Muc2 in the intestine). It then argues that MUC5AC is the principal gastric gel-forming mucin, making KLF4-driven regulation biologically plausible. By marrying KLF4's differentiation role with the barrier function of MUC5AC, the reasoning shows a nuanced grasp of gastric physiology. |

---

**SCPILOT Prompt for Cell type annotation - hypothesis**

content = f"Top {len(self.top_genes)} differentially expressed genes: {self.top_genes}"
if self.reference_dict:
content += f"You can refer to the possible cell types of these top genes in this dictionary-self.reference_dict"
content += f"Current Hypothesis:self.hypothesis"
if annotation_dict:
content += f"The cell type annotation from previous iterations annotation_dict"
if no_gene_cluster: content += f"Clusters without need to be focused on: {no_gene_cluster}"
if iteration_summary:
content += f"This is summary of previous iteration annotation, with information of next steps to take. iteration_summary"
system_role = "You are a research assistant specializing in cell biology. Based on top differentially expressed genes, previous cell type annotation (if provided), Clusters without need to be focused on (if provided), summary of previous iteration annotation (if provided), and failed genes (if provided), refine the given hypothesis to be more accurate and specific."

---

In the marker gene proposal step, SCPILOT specifically proposes a marker gene list for the cell types of interest.

---

**SCPILOT Prompt for Cell type annotation - marker gene proposal**

prompt = f"' You are a bioinformatics expert specializing in liver cell annotation. Your task is to propose an experiment for cell type annotation based on the following information:
Refined hypothesis: self.hypothesis

---

Table 24: SCPILOT o1 failed reasoning in TF-Gene Prediction

| TF → Gene (Tissue) | Reasoning | Why the reasoning failed |
|---|---|---|
| Usf2 → Pigr (Stomach) | Usf2 is a bHLH-LZ transcription factor best characterized for regulating metabolic and some housekeeping genes. Pigr encodes an immunoglobulin receptor primarily active in mucosal immunity.

There is no known functional overlap or evidence of co-regulation between Usf2 and Pigr in stomach tissues. Hence, a direct Usf2 → Pigr regulation in the stomach is unlikely. | The model claimed 'no known functional overlap' because PIGR is a mucosal-immunity gene and USF2 'mostly regulates metabolic genes'. In reality, USF family bHLH factors bind canonical E-box motifs present in the Pigr promoter and are documented activators of several immune-related transporters. By restricting USF2 to 'metabolic housekeeping', the explanation ignored promoter architecture studies that tie USF2 to Ig-transport regulation in gut epithelium—hence the false negative. Actually, one published journal [**?** ] has explicitly mentioned this TF-gene relationship, indicating that the current SCPILOT is not an expert in searching literature. |
| Fos → Hmox1 (Stomach) | Fos is a key component of the AP-1 complex, known to regulate stress-responsive genes, including those involved in cellular response to cadmium ion. Hmox1 is upregulated by oxidative and heavy metal stress and shares this functional pathway with Fos targets. This overlap suggests moderate potential for direct regulation of Hmox1 by Fos in the stomach. | The model argued that both genes are "stress-responsive" and share the GO term cellular response to cadmium ion, concluding a plausible AP-1–driven induction. HMox1 transcription is, in fact, dominated by NRF2/KEAP1 and small-Maf binding at antioxidant-response elements, not by AP-1 sites. While AP-1 motifs exist in the far upstream enhancer, they are not functional in gastric tissue for HMox1. The explanation, therefore, over-generalized "stress overlap" without checking the principal regulatory axis, yielding a false positive. |

Instructions:
0. We have already labeled some of clusters, the information is in annotation_dict
1. a) The most important unsolved clusters are no_gene_cluster.
b) Other cell types in annotation_dict is cell types we labeled with high or low confidence. We first need to decide whether we should annotate them again.
2. For the cell types we have already successfully used, the gene marker list is successful_genes.
For large cell types like hepatocyte or B cell, the remaining clusters might still contain them, but for smaller cell types you don't need to think about them again. For the marker genes we already used but did not detect any expression, the list is failed_genes. So you don't need to try these gene and related cell type again.
3. Do not specify the cluster with cell type here. You can just output cell type and related marker genes.
4. Only provide proposal of unlabeled clusters, consider potential overlaps in marker gene expression between cell types:
a) Name of the cell type b) 3-5 marker genes
Output format: 1. List of cell types with their markers: [Cell Type 1]: Gene A; Gene B; Gene C [Cell Type 2] ...
2. Python list of all marker genes: MARKER_GENES = ['GeneA', 'GeneB', 'GeneC', ...]
Remember: - Be specific and concise in your descriptions. - Ensure all cell types have at least 3 marker genes. - Include the Python list of all marker genes at the end of your response, don't use any backtick. '''

system_role = "You are an AI trained to design scientific experiments based on hypotheses and background information."

In the evaluation step, SCPILOT analyze the dotplot expression data, and make comprehensive evaluation about the expression and thus perform prediction.

---

**SCPILOT Prompt for Cell type annotation - evaluation**

content = f'" Context: You are working on a single-cell RNA sequencing cell type annotation task. The goal is to identify distinct cell types based on gene expression patterns. You have created a dotplot using a set of marker genes to visualize gene expression across different clusters.

MUST remember: you should list the cell types here. All cell types you refer to, should be in here. possible_cell_types

Data: For each cluster, the top genes are in this dictionary: marker_genes The clusters cannot find top genes are: empty_keys These are the genes that are successfully (highly) expressed in some clusters: success_list These are the genes that are failed to express in any clusters: fail_list There are some clusters that have similar expression, so we did differential expression analysis for these cluster pairs. The pairs and top differential genes are in: similar_clusters_dict

Add-on: Duplet and Contamination in dotplot: duplet_rule Any other important instructions: contamination_rule

Please give greatest attention to the Add-on part, if they are provided. Make sure to use them in your analysis.

Instructions: Please analyze the provided data and answer the following questions:

1. Gene-level analysis: a) Which genes are highly expressed in specific clusters? Provide a detailed description. b) Are there any genes that show differential expression across clusters (high in some, low in others)? c) Are there any genes that are not informative for cell type annotation (low expression across all clusters)? You should answer this question based on negation of 1b.

2. Cluster-level analysis: a) Are there any clusters that lack high expression of any marker gene? If so, list the cluster numbers. There are in total cluster_size clusters, index from 0.

3. Overall assessment: a) Based on the gene expression patterns, are there distinct clusters that potentially represent different cell types? b) Can you assign specific cell type identities to any of the clusters based on the marker gene expression? If so, provide your cell type annotations. You should only assign one cell type to each cluster. No doublet or contamination here. c) To refine the cell type annotation, recommend possible additional cell types. d) To refine the cell type annotation, recommend any particular cluster to perform subgrouping.

4. Confidence assessment: a) What are confidence levels of your annotation? Please also assign a confidence score to the process name you selected. This score should follow the name in parentheses and range from 0.00 to 1.00. A score of 0.00 indicates the lowest confidence, while 1.00 reflects the highest confidence. This score helps gauge how accurately the annotation is. Your choices of confidence score should be a normal distribution (average = 0.5) You should consider if the annotation is widely seen, using deterministic wording and following background of dataset. For instance, if you label a doublet, or using word "probable", the score should be lower.

b) Based on confidence levels, what are the annotation results that you want to "stabilize", that is not change in next steps? if iteration is 1, you should choose top 1/3 confident clusters. if iteration is 2 and beyond, you should choose top 2/3 or more clusters. You can choose this threshold.

Please provide your answers in a structured JSON format, addressing each question separately using "1a" "2a" etc.

Remember, this is one iteration cell type annotation for a liver scRNA-seq dataset. Your insights will guide further refinement of the analysis. '" system_role = "You are an expert bioinformatician specializing in single-cell RNA sequencing data analysis and cell type annotation."

## E.3 Trajectory Inference

Both Direct prompting and SCPILOT use the same simple annotation prompt to generate annotated clusters for trajectory analysis.

---

**Direct and SCPILOT Prompt for Trajectory - annotate**

query = f""" You are analyzing single-cell RNA-seq data. The dataset contains clusters of cells identified by Leiden clustering. Given these clusters and reference cell types, predict the most likely annotation for each cluster. First, read the context. Then, use the top 5 genes to annotate each cluster. Finally, refine the annotation with the percentage of timepoint in each cluster, so you will know the cell type in the cluster is more likely in the proliferating or mature stage.
Please refer to the context information of the dataset: context Please base your annotation on the top 5 genes of each cluster: top5_dict Here is the percentage of timepoint in each cluster: day_percentage
Remember, every cluster is a distinct cell type.
First output your chain of thought, then provide output as a dictionary mapping cluster IDs to cell type annotations. For the dictionary, ONLY output a python code dictionary, do not include "'python "'. """

---

The Direct Prompting use a one-step tree construction prompt, trying to connect all annotated clusters in one tree.

---

**Direct Prompt for Trajectory - build tree**

query = f""" Construct a developmental trajectory tree for the clusters in the single-cell dataset. Here is the context: context Ensure: 1. The tree starts from the root (youngest stage). 2. Progression follows biologically meaningful paths. 3. Misplaced branches are flagged.
Data: 4. Cell Type annotation for each cluster: annotated_clusters 5. Developmental stages: day_percentage
In your trajectory tree, do not include any time stage, only use the Cell clusters. We need to set a dummy node as root (use the name "root"), and then add all cell types iteratively as leaves and leaves of leaves. DO NOT include any other node except root and cell types.
First output your chain of thought, then provide output trajectory tree as a nested dictionary at the END. For the nested dictionary, ONLY output a python code dictionary, no backslash n to make new line. do not include "'python "'. Do not put anything else after the nested dict. """

---

SCPILOT adopts a multi-step tree construction prompting. First, find the root node (initial cell type). Then iteratively add the leaves to the tree and then finalize the trajectory.

---

**SCPILOT Prompt for Trajectory - multi-step tree reconstruction**

root_query = f""" You are analyzing a single-cell RNA-seq dataset with developmental progression. Context: context
Given the following: - Cell Type annotation for each cluster: annotated_clusters - Developmental stages as percentage across timepoints: day_percentage
Task: Identify the single most appropriate root cluster for a developmental trajectory tree. This should represent the developmentally earliest or least differentiated cell population.
Only return the name of the root cluster as a string. Do not include any additional explanation. """

tree_query = f""" Construct a developmental trajectory tree starting from the root cluster: "root_cluster".
Context: context
Given: - Cell Type annotation for each cluster: annotated_clusters - Developmental stages: day_percentage
Task: Iteratively construct a trajectory tree: 1. The tree should begin at the root cluster: "root_cluster". 2. Add directional edges from the root to other clusters based on develop-

---

mental progression. 3. Each edge should represent a biologically meaningful transition. 4. Continue expanding the tree until all clusters are connected. 5. Flag any biologically implausible transitions (e.g., backward differentiation).

Output Format: Return a Python dictionary where: - Keys are parent nodes (clusters). - Values are lists of child clusters. Ensure every cluster appears at least once in the tree.

In your trajectory tree, do not include any time stage, only use the Cell clusters. We need to set a dummy node as root (use the name "root"), and then add all cell types iteratively as leaves and leaves of leaves. DO NOT include any other node except root and cell types. """

finalize_query = f""" Please extract the trajectory tree inside the input. The input containing the tree is whole_tree IN YOUR REPLY, ONLY output python code tree, DO NOT include "'python "'. You can represent a tree using nested dict. DO NOT USE any square brackets. ALWAYS use nested curly brackets.

---

After receiving the report from py-Monocle, SCPILOT will generate an analysis summarizing how to improve based on Monocle suggestions.

---

**SCPILOT Prompt for Trajectory - analysis monocle**

analysis_prompt = f""" Analyze these components for trajectory improvement: 1. Biological context: context 2. Current trajectory: trajectory_tree 3. Cluster annotations: annotated_clusters 4. Analytical report: trajectory_report Identify: - Missing progenitor relationships - Cluster-cell type mismatches - Marker gene contradictions - Structural hierarchy errors - Root node completeness """

---

Then, SCPILOT will reconsider the cell type annotation and trajectory with the Monocle analysis.

---

**SCPILOT Prompt for Trajectory - reconsider with monocle**

query = f""" Perform comprehensive validation of cell trajectory and annotations using biological context and analytical report insights.

Requirements: 1. Analyze context to identify key gene markers for cell fate determination. 2. Validate hierarchical structure in trajectory_tree matches differentiation pathways. 3. Resolve any inconsistencies in annotated_clusters using markers from trajectory_report. 4. Ensure root node contains all initial progenitor states. 5. Remove temporal references, focus on lineage relationships.

Current trajectory structure: trajectory_tree

Existing cluster annotations: annotated_clusters

This is the analysis about how to improve: analysis

No matter how you modify, you should make sure all the cell types in the final cell type annotation dict are in the final trajectory structure, and vice versa.

Output Format requirement: - DO NOT use "'python"' to wrap your python code. - Return as string of two dictionaries: trajectory_dict and annotation_dict. - Trajectory must be hierarchical dict. set a dummy node as root (use the name "root"), and then add all cell types as leaves. - In your trajectory, do not include any square brackets. use 'str' to contain nodes, and curly bracket to show tree trajectories. - In the trajectory, Leaf node values should always be empty dictionaries (empty curly brackets) - Annotation dict must map ALL cluster IDs to terminal cell types.

ONLY output the string of two dictionaries, no additional text/formatting. Be extra careful with the format of curly brackets in the nested dict of trajectory_dict. """

---

SCPILOT will have a final synthesis step that keep all the reconsidered results consistent.

---

**SCPILOT Prompt for Trajectory - final synthesis**

synthesis_prompt = f""" Current adjusted trajectory and refined annotations: response
You may see it is a string containing one nested dictionary for trajectory, and one dictionary for annotation. We want to extract these dictionary from the string.

---

> Make sure to output the current input materials in correct format: Make sure all the cell types in the final cell type annotation dict are in the final trajectory structure, and vice versa.
> Output Format requirement: - DO NOT use "'python'" to wrap your python code. - Return as a single Python tuple with (trajectory_dict, annotation_dict) - Trajectory must be hierarchical dict. set a dummy node as root (use the name "root"), and then add all cell types as leaves. - In your trajectory, do not include any square brackets. use 'str' to contain nodes, and curly bracket to show tree trajectories. - In the trajectory, Leaf node values should always be empty dictionaries (empty curly brackets) - Annotation dict must map ALL cluster IDs to terminal cell types
> ONLY output the Python tuple (trajectory_dict, annotation_dict), no additional text/formatting.
> """

For GRN TF-gene prediction, the Direct Prompting only asks a simple question regarding this TF-gene relationship.

---

**Direct Prompt for GRN prediction**

prompt = f""" Decide how much possible tf directly regulates gene in (ctxB):
The possibility is a number from 0 to 1.
Return exactly: Reasoning: <your reasoning> Possibility is: <your possibility> """.strip()

---

For SCPILOT, the GRN TF-gene prediction incorporates more inputs. There is more context about the tissue of TF and gene, and the GO database functional overlap for TF and gene.

---

**SCPILOT Prompt for GRN prediction**

prompt = few_shot_block() + f""" *Task*: • TF: tf and Context A tissues (ctxA) • Functional overlap (shared GO BP terms): overlap
Decide how much possible tf directly regulates gene in (ctxB):
The possibility is a number from 0 to 1.
Think step by step: 1. Recall TF tf's biological role. 2. Compare gene with known tf targets. 3. Conclude which statement fits better (<= 4 sentences).
Return exactly: Reasoning: <your reasoning> Possibility is: <your possibility> """.strip()

---

